# Degradability and Properties of PBAT-Based Biodegradable Mulch Films in Field and Their Effects on Cotton Planting

**DOI:** 10.3390/polym14153157

**Published:** 2022-08-02

**Authors:** Qi Liu, Yue Wang, Jialei Liu, Xiaowei Liu, Yanyan Dong, Xin Huang, Zhichao Zhen, Jun Lv, Wenqing He

**Affiliations:** 1National Engineering Laboratory for Crop Efficient Water Use and Disaster Mitigation, Key Laboratory of Prevention and Control of Residual Pollution in Agricultural Film, Ministry of Agriculture and Rural Affairs, Institute of Environment and Sustainable Development in Agriculture, Chinese Academy of Agricultural Sciences, Beijing 100081, China; liuqi@caas.cn (Q.L.); wyw111y@sina.com (Y.W.); liujialei@caas.cn (J.L.); dongyanyan@caas.cn (Y.D.); huangxinharris@sina.com (X.H.); 2Shihezi Agricultural Science Research Institute, Shihezi 832000, China; liuxiaowei1212@sina.com; 3Technical Institute of Physics and Chemistry, Chinese Academy of Sciences, Beijing 100045, China

**Keywords:** PBAT, agricultural mulch, degradation, duration, yield

## Abstract

Biodegradable mulches (BDMs) are considered promising alternative green materials to achieve the substitution of polythene (PE) films to reduce plastic pollution. However, whether the BDMs are sufficiently effective to promote cotton production as PE film is a controversial topic. In this study, laboratory determination and field experiments were conducted with one pure Poly(butylene adipate-co-terephthalate) (PBAT) film (BDM), two commercial PBAT-based films (BDM1 and BDM2), and one PE film to (ⅰ) compare the degradation behavior, morphology, and property changes during field application, and (ⅱ) reveal their effects on biomass accumulation and cotton yield. Degradation behavior, degradation rate, structure, thermal stability, crystallinity, and molecular weight changes of the films before and after mulching were investigated and characterized. Water vapor transmission rate and mechanical properties of the films and the effects these on soil temperature, crop growth, and cotton yield were discussed. Results show that the three PBAT-based mulch films gradually degraded during mulching. The molecular weight, thermal stability, and crystallinity of BDM1 and BDM2 decreased. Interestingly, BDM showed the opposite characteristics, but the degradation degree was greatest at harvest. PE film showed no significant changes in its microscopic appearance, thermal performance, or properties. These PBAT-based films were positively correlated with the complete coverage period of the films. In-depth studies focused on BDMs with a longer mulching period must be developed to promote the substitution of BDMs into PE to reduce the residual mulch pollution in cotton fields.

## 1. Introduction

Biodegradable mulches (BDMs) may represent an effective green alternative to reduce plastic residue risks in farmland [1,2]. These kinds of materials have shown disintegration and mineralization comparable with paper and leaves and compliance with the European standard EN 17033 [3,4]. BDMs have been tested and evaluated on various crop planting, such as tomato, strawberry, maize, and potato, all over the world [5,6,7,8,9,10]. China accounts for approximately 30% of the world’s cotton output with 15% of the world’s cotton land [11]. However, poor growing conditions in most plating regions, such as the shortage of water resources and the low temperature between April and May, make mulching film a necessary measure for cotton planting. Nevertheless, the extensive use of mulch film and the lack of recycling have led to serious residual film pollution. Among all crops, cotton fields yielded the largest residual amount of mulch film (158.4 kg·ha^−1^), in addition to making the largest contribution (2.6 × 10^5^ tons) to the total amount of residual film in Xinjiang, China, in an investigation in 2013 [12].

In recent years, biodegradable polymers have gradually become the main raw material of BDMs [13,14,15,16,17]. Among these, poly(butylene adipate-co-terephthlate) (PBAT) and its composite films are the most popular and important ones [18,19]. Numerous studies focus on the possibility of utilizing BDMs for cotton plating. The effects of PBAT-based BDMs on soil temperature and moisture, water use efficiency (WUE), and yield are widely explored [20,21,22]. Deng et al. [20] investigated the substitution of common PE film with PBAT BDMs in the cotton production in Xinjiang and found that BDMs films had comparable results to PE one in increasing crop growth, yield, and WUE, which was preferable for agricultural applications in this area and supported the sustainable development of agroecosystems. Wang et al. [21,22] evaluated different BDMs as alternatives of PE films for cotton planting in Xinjiang and found no significant difference between yields with PE films and those with thicker PBAT films (12 μm), which were suggested as potential alternatives to PE films for cotton planting and for controlling soil pollution under mulched drip irrigation.

However, the results of cotton yields among different studies for BDMs have not had consistent results [20,21,22,23,24,25]. Some studies found that BDMs have similar results as PE film in increasing crop growth and yield [20,21,22], but others found that BDMs cannot replace PE film in cotton planting because it reduces cotton yield [23,24,25]. Wu et al. carried out a field tests in 2015 and 2016 and found that the BMDs decreased the cotton yield of the two years by 2.89% on average (*p* < 0.05) and reduced the WUE by about 4% compared with PE film [23]. Dou et al. compared the effects of different degradable mulches (all 10-μm thickness) on cotton growth, yield, and economic benefits in southern Xinjiang and found that the seed cotton yield of BDMs decreased by 30.26% compared with that of PE film [24]. The most probable cause of this problem is the inaccurate thickness of the mulch film used in these studies, and all the thickness values of the employed BDMs were given by the producer instead of experimental tests. Nevertheless, thickness has been proven to have a crucial impact on the effective mulch duration [21,22]. Therefore, to address this issue, one important purpose of this study is to evaluate the alternative effects of BDMs on PE film in cotton planting based on the accurate preparation and measurement of the thickness of the mulch films.

Furthermore, the structural and property changes during the degradation process in the field have rarely been studied in previous works. Therefore, the degradation behavior, composition, and structural changes of BDMs in the field are not characterized and described well. The relationship between its degradation speed and cotton yield is also not documented. Therefore, another important aim of this work is to investigate the degradation behavior, degradation rate, structure, and performance characteristics of BDMs, as well as cotton growth and yield during field tests to reveal the effects of BDMs’ degradation rate as well as changes in the properties and structure on cotton growth and yield.

In this study, one PBAT film and two commercial PBAT-based ones were prepared at the same thickness for comparison. The three BDMs were tested for cotton planting by taking PE film as control. The degradation behavior and degradation rate were investigated through visual observation, digital photo, and scanning electron microscopy (SEM). Fourier transform infrared (FTIR) spectroscopy, thermogravimetric (TG) analysis, and gel permeation chromatography (GPC) were conducted to analyze the structure, thermal stability, crystallinity, and molecular weight change of the films, respectively. The water vapor transmission rate (WVTR) and mechanical properties were tested to evaluate the function during the growing season. Effects on soil temperature, crop growth, and cotton yield were discussed.

## 2. Materials & Methods

### 2.1. Materials

PBAT was purchased from JinhuiZhaolong Co., Ltd. (Lvliang, China) with the trade name of Ecoworld^®^. It possessed a density of 1.26 g/cm^3^, a melt index of 3–5 g/10 min (190 °C/2.16 kg), and an average melting temperature of 115 °C. For comparison, two commercial PBAT-based master batch for making BDMs were also bought from BASF company (Ecoflex^®^, Ludwigshafen, Germany) and Kingfa Science and Technology Co. Ltd. (Guangzhou, China, FLEX-262 F20). *N,N*-ethylene Bis-stearamide (EBS, >98 wt%) was obtained from the Shanghai Macklin Biochemical Co. Ltd. (Shanghai, China).

### 2.2. Preparation of PBAT Films

PBAT films were dried in a vacuum oven at 80 °C for 24 h before use. Then, PBAT (JinhuiZhaolong Co., Ltd., Lvliang, China) was mixed with 0.5 % EBS in a plastic mixer. The compounds were prepared by melt-mixing in a twin-screw extruder (LTE-26-44, Labtech Engineering, Samut Prakan, Thailand) followed by processing into pellets by a pelletizer cutting machine (LZ-120/vs, Labtech Engineering, Samut Prakan, Thailand). The temperature profiles were set from the feed zone to die around 170–185 °C. The feed speed and screw speed were 20 rpm and 100 rpm, respectively. The extrudate was cooled in a water bath, pelletized, and dried at 60 °C for 6 h. After that, the PBAT films were prepared by a blown film extruder (QY2200-1, Shandong Plastic Machinery Company, Jining, China) at 135 °C, naming as BDM. Then, another two films were prepared through the same blown process conditions directly using commercial PBAT-based master batch Ecoflex^®^ and FLEX-262 F20, which were named as BDM1 and BDM2, respectively. The blow-up ratio of the bubble was 3.5:1. This setting produced a bubble with an average thickness of 10.0 ± 1.0 μm (Appendix A).

### 2.3. Field Tests

#### 2.3.1. Site Description

Field tests were conducted in Shihezi (43°26′–45°20′ N, 84°58′–86°24′ E), Xinjiang, in 2018. The study site has a typical temperate continental climate. The average annual sunshine duration amounts up to 2669 h, the annual average temperature is 7 °C, and the frost-free period lasts 160 days.

The soil at this site is medium loam with a pH of 7.9. The average field water holding capacity is 0.30 (*m*/*m* of soil), and the average soil bulk density is 1.38 g/cm^3^. The soil bulk density, total organic matter, available N, available P, and available K within 0–30 cm topsoil were measured in the laboratory (Appendix A). The study site was planted with cotton in 2016 and 2017.

#### 2.3.2. Experimental Design

Mulching drip irrigation was adopted for cotton planting. Six rows of plants were sown under one strip of mulch (205 cm wide) with three drip tapes. Dibble sowing, which allowed drip tape laying, mulching, and seed sowing to be completed in one run was carried out. Cotton (*Gossypium hirsutum* L.) was sown at a rate of 240,000 plants ha^−1^ on 23 April 2018. Each treatment had three replicates, and each plot was 50 m × 4.5 m. In total, 12 plots, that is, four treatments (three covered by different BDMs, one by PE film as control) × 3 replicates were established. All the other agriculture managements were kept the same with local cotton fields. The first drip-irrigation was set on 24 April.

After several days, three specimens of each sample were sampled to obtain a sample sufficient for characterization and properties tests. The BDM, BDM1, and BDM2 films after 165 days mulching in the field were named BDM’, BDM1’, and BDM2’, correspondingly.

#### 2.3.3. Quantitative Degradation Grading Observation

After mulching, degradation investigation of the mulch in the field was carried out every 1–2 weeks, and the frequency of investigation in the early degradation stage was increased. Five 0.5 m × 5 m areas that were selected for each treatment were investigated (Table 1). The degradation degree was divided into Level 0 to Level 6 and defined as follows:

Level 0: The surface of the mulch is completed with almost no visible cracks or holes. Level 1: More than 5 holes with a diameter greater than 2 cm or (and) cracks with length greater than 5 cm are visible on the mulch film per m^2^. Level 2: More than 5 holes with a diameter greater than 10 cm or (and) cracks with length greater than 50 cm are visible on the mulch film per m^2^. Level 3: More than 5 holes with a diameter greater than 20 cm or (and) cracks with length greater than 100 cm are visible on the mulch film per m^2^. Level 4: More than 5 holes with a diameter greater than 30 cm or (and) cracks with length greater than 200 cm are visible on the mulch film per m^2^. Level 5: The surface of the mulch almost fragmented, and no fragment bigger than 0.5 m^2^ is visible. Level 6: The surface of the mulch almost completed degradation, and no fragment bigger than 0.05 m^2^ is visible.

#### 2.3.4. Agronomic Performance Assessment

Soil temperature change along time was monitored by means of probes placed at 15 cm depth into the soil simultaneously with the installation of the films.

The emergence rate was measured 10 days after emergence. Three sampling areas were selected for each treatment with diagonal sampling method, and 100 holes of each area were investigated. The emergence rate was calculated by dividing the number of emergence holes by the number of surveyed ones.

The canopy closure was determined by aLAI-2000 Plant Canopy Analyzer (LI-COR, Lincoln, NE, USA) 8 weeks after the installation of the films and was performed every 2 weeks according to the growth condition of the crop.

Plant height and effective boll of each plant were measured on 20 July, one day before tip pruning of the cotton. The number of cotton plants and bolls in a 9.2 m^2^ sample area was investigated in each test plot.

Weeds under the completed mulching film of each plot were collected, cleaned, dried, and weighted on 20 June and 20 July. The weed inhibition rate of each film was calculated as the total of weight percentage by dry weed weight of each to that of the non-mulching treatments.

A total of 100 cottons in the plot were collected to measure the boll weight to calculate cotton yield for each plot on 13 October.

### 2.4. Characterization and Properties Tests

#### 2.4.1. Imaging of Sample Surfaces

Macroscopic surface changes of the films were observed by digital camera in a square with a side length of 50 cm as well as along a line of mulched cotton, while microscopic changes were analyzed by a scanning electron microscope (SEM). The morphology of BDMs used for films mulched for 0, 87, and 165 days were visualized by SEM (Hitachi S-4800, Hitachi High Technologies America, Inc., Tokyo, Japan). Samples were cut into 4 mm wide, coated with gold by an ion sputter coater, and then observed under SEM at 15 kV.

#### 2.4.2. FTIR Spectroscopy

FTIR spectroscopy was adopted to examine the structural changes between BDMs before and after use. The spectrum of each sample was recorded with a Micro-Attenuated Total Reflection Fourier Transform Infrared (Micro-ATR-FTIR) microscope (OPUS 7.5, Bruker, Germany). The FTIR spectrum of each sample was obtained at 4000–400 cm^−1^ with a resolution of 4 cm^−1^.

#### 2.4.3. TG Analysis

TG analysis was carried out using a TGA/DSC 1 thermogravimetric analyzer (Mettler Toledo Corporation, Zurich, Switzerland) within a temperature range from room temperature to 800 °C at a heating rate of 10 °C/min under nitrogen atmosphere. A total of 5–10 mg sample was tested for films before and after mulching in the field.

#### 2.4.4. Differential Scanning Calorimetry (DSC)

DSC (Q200, TA instruments, New Castle, DE, USA) was carried out to describe the melting behavior and crystallinity of the films before and after mulching in the field. A total of 5–10 mg sample was weighed and placed in a crucible under a nitrogen atmosphere. A three-step procedure of heating/cooling/heating ramps was adopted. The temperature program was set as follows: Samples were heated from room temperature to 200 °C at 10 °C/min and kept at a constant temperature for 3 min to eliminate the thermal history. Then, subsequent cooling was performed from 200 °C to room temperature at 10 °C/min. In the second heating run, the sample was heated from room temperature to 200 °C at 10 °C/min again to obtain the second heating curve. The relative crystallinity ($X_{C}$) is calculated using Equation (1)
(1)$$X_{C}=∆H_{m}/\left(\omega \times ∆H_{m}^{*}\right)\times 100\%$$
where $∆H_{m}$ is the melting enthalpy detected in the experiment; ω is the weight fraction of PBAT in the blend; and $∆H_{m}^{*}$ represents the perfect enthalpy of 100% crystalline PBAT, which is calculated as 114 J/g [26]. The percentage of crystallinity increasing amount (θ) after mulching can be calculated by Equation (2)
(2)$$\theta =\left(∆H_{m}^{f}-∆H_{m}^{0}\right)/∆H_{m}^{0}\times 100\%$$
where $∆H_{m}^{0}$ is the melting enthalpy detected of the sample before mulching and $∆H_{m}^{f}$ is the one after mulching.

#### 2.4.5. GPC

GPC was used to determine the number-averaged molecular weight (M_n_), weight average molecular weight (M_w_), and polydispersity index (M_w_/M_n_, PDI) of the films before and after mulching in the field. Degradable samples were dissolved in chloroform, which were then centrifuged and passed through 0.20 μm nylon filters to remove insoluble materials. The filtered solution was determined by gel permeation chromatography (LC20, Shimadzu, Kyoto, Japan) at 30 °C, tetrahydrofuran was used as mobile phase at a flow rate of 1.0 mL·min^−1^, and calibration was performed using polystyrene (PS) standards. PE samples were determined by a high-temperature GPC (HT-GPC), the measurements of which were determined with a HT-GPC (PL-GPC220, Agilent Technologies, Santa Clara, CA, USA) with three PLgel MIXED-B LS 300 × 7.5 mm columns. TCB stabilized with 0.0125% BHT was used as eluent at a flow rate of 1.0 mL·min^−1^ and temperature 150 °C. The instrument was also calibrated with PS standards.

#### 2.4.6. Water Vapor Permeability Testing

To determine the moisture permeability of the films, WVTR were determined by a water vapor transmission rate tester (W3/060 PERME, Labthink, Jinan, China) according to ASTM E96/E96M-2014 [27]. The BDM samples were cut using a sampler with an area of 33.2 cm^2^. Samples were sealed on top of the permeation cells with appropriate amount of ultra-pure water. The cells were set in the sample chamber of the water vapor transmission tester. The pressure of the output pressure gauge is 4–5 MPa, and the pressure of the automatic drying filter is 0.3–0.35 MPa by adjusting the air source of the system. The temperature and relative humidity of the tester were set to 38 °C and 90% RH. All measurements were repeated three times.

#### 2.4.7. Mechanical Property Testing

A universal testing machine (XLW, Labthink, Jinan, China) was used to determine the tensile and tear properties of the BDMs. Tensile properties were determined according to the ASTM D882-2018 method [28]. Films were cut into 10 mm wide strips and at least 80 mm long. The grip separation and crosshead speed were 50 mm and 500 mm/min, respectively. At least five specimens were tested for each sample. The tear properties of films were determined according to the ASTM D1004-2013 method [29]. The samples were cut into the shape of a mold. The strain rate was 200 mm/min. At least five specimens were tested for each sample. The thicknesses of five replicates of each sample were measured with a micrometer (C1200, Mahr Millimar, Esslingen, Germany) with an accuracy of ±0.1 µm according to the ASTM D6988-2013 method [30].

## 3. Result & Discussion

### 3.1. Morphology

To compare the morphological changes, a group of scanning electron microscope (SEM) images of BDM, BDM1, BDM2, and PE before, during, and after mulching are shown in Figure 1. Before mulching (at Day 0), all four samples were complete without any obvious holes or splitting. The surface of BDM was more uniform than those of BDM1 and BDM2. Different size “hills” could be observed on surface of BDM1, but smaller particle with diameter of 1–3 μm was observed on the surface of BDM2. The structure of PE was very dense with some fine texture. After 87 days of mulching, some degradation trace as tiny tears and holes can be found on the surface of BDM. However, it became more uniform and denser. There were more small particles and tiny holes on the surface of BDM1. Meanwhile, the surface of BDM2 shows obvious degradation phenomenon and large cracks. More large and small particles appeared on the surface, which were more prominent on the surface. No obvious changes were observed in PE. After 165 days, all the three PBAT-based samples degraded further. The surfaces of BDM and BDM1 were similar, on which large cracks can be seen. However, the overall structure was not obviously loose, and the structure of BDM2 loosened further, with many obvious holes. Still, no obvious changes were observed in PE. Therefore, all PBAT films degraded in the farmland environment with different degradation behavior because of their different components.

### 3.2. Structure

Figure 2 compares the infrared spectra of various films before and after aging. Nearly all spectra showed PBAT characteristic peaks, except those of PE mulch films, which presented special characteristic peaks. In Figure 2a–c, the infrared spectral information showed similarities: the absorption peaks near 2930–2950 cm^−1^ correspond to the stretching vibration peak of C–H in PBAT; the absorption peak at approximately 1710 cm^−1^ should be assigned as the stretching vibration of C=O in PBAT; the absorption peak from 1270–1010 cm^−1^ were all stretching absorptions of C–O–C, which were complex because of the different atoms of function groups adjoining to it; and the absorption peaks near 728–727 cm^−1^ represent the bending vibration absorption of C–H of the benzene ring. In Figure 2d, the absorption peaks near 2920 cm^−1^ and 2850 cm^−1^ correspond to the asymmetric stretching vibration and the symmetric stretching vibration of C–H in PE, respectively. The absorption peak at 1470 cm^−1^ was the bending vibration peak of C–H in PE. The absorption peak at 718 cm^−1^ presented the rocking peak of C–H in PE.

### 3.3. Molecular Weight

Figure 3 compares the molecular weight distribution curves of various films before and after aging in the field. M_n_, M_w_, and the resulting PDI of each sample are listed in Table 2. As shown in Figure 3a, the molecular weight distribution curve of BDM moved slightly to a higher molecular weight after aging, which was possibly caused by partly chain cross-linking under ultraviolet radiation in the field. M_n_ and M_w_ increased, and M_w_/M_n_ decreased possibly be due to the simultaneous obvious occurrence of degradation and crosslinking in BDM. For BDM1 and BDM2, the values of M_w_ and M_w_/M_n_ decreased with the increase in M_n_ after aging, indicating that these two mulch films degraded obviously and produced a large amount of low molecular-weight substances. The molecular weight distribution curves of BDM1 and BDM2 shifted to the left after aging obviously, indicating that the two films have severe chain scission and degradation. A 15.0% and 52.6% loss of M_w_ were observed. The molecular weight distribution curve of the PE mulch was almost unchanged after aging, thereby manifesting that it did not degrade significantly.

### 3.4. Thermal Analysis

TG and differential scanning calorimetry (DSC) of samples before and after mulching were carried out for thermal analysis. Because BDM1 and BDM 2 were blending materials with mostly PBAT and some other polymers, we only discuss the thermal behavior of PBAT in this study. The results of TG determination are shown as TG curves and derivative thermogravimetric (DTG) curves (Appendix A). The DSC cooling and heating curves of samples before and after mulching are shown in Figure 4. The thermal data extracted from these thermograms are presented in Table 2. Only one peak was observed in the DTG curve of BDM. Meanwhile, approximately 10% thermal weight loss was observed at 327.5 °C and 324.9 °C for BDM1 and BDM2 (Appendix A), respectively. These findings demonstrated that there was another kind of polymer in these two films. After 165 days of mulching, the DTG curve of BDM 1 and BDM 2 migrated to a lower temperature for 1.6 °C and 4.4 °C, whereas no obvious change to that of BDM was noted, indicating that the TG temperature of the corresponding PBAT decrease in these two samples after degradation. Almost no change was observed for PE.

As shown in Figure 4 and Table 2, the crystallization and melting temperatures of the BDM sample decreased after mulching in the field, where BDM2 showed the most significant decline. The crystallization and melting temperatures of BDM2 were 87.5 °C and 122.2 °C, respectively, at the beginning. In the end of mulching, these temperatures changed to 77.2 °C and 117.3 °C. A slight or not significant change was observed in the *T*_c_, Δ*H*_c_, *T*_m_, and Δ*H*_m_ of PE film after field aging. The results of percentage of increased crystallinity of PBAT, *Θ*, were calculated through Equation (2) assuming that the percentage composition of PBAT in the sample before and after mulching are the same. The results showed that the crystallinity of PBAT increased in sample BDM’ while it decreased in BDM1′ and BDM2′, indicating the occurrence of crosslinking in sample BDM during the mulching period. A 59.3% decrease of *Θ* of BDM2 demonstrated its serious degradation during the mulching period. This result was consistent with those of SEM and GPC.

### 3.5. Functional Performance for Mulching

The most important functions of mulching films are to increase temperature, conserve moisture, and prevent weeds. Data for variation in the time of tensile strength, WVTR, and weed suppression are presented in Figure 5. The tensile strength results of the four samples at various mulching periods are shown in Figure 5a. BDM and BDM1 showed similar trends, in which the tensile strength increased in the late mulching period, especially for BDM. This phenomenon may be caused by the crosslinking characterized by SEM, GPC, and DSC. BDM2 became stronger around 40 days and weakened immediately after that. No complete fragments for mechanical tests could be collected after 120 days of mulching. No significant changes in tensile strength were observed for PE during the growing season.

Increasing soil temperature is one of the most important functions of mulching films. The recorded soil temperature is shown in Figure 5b. As seen from Figure 5b, the soil covered by PE achieved the highest temperature during the first 75 days after mulching. Subsequently, no significant difference was observed among the four treatments. When compared with the three degradable films during the first 35 days, BDM2 reached the highest temperature. However, it became lower than the other two PBAT-based films after that, especially for the duration between 40 and 60 days, during which BDM2 underwent obvious degradation and accelerated degradation periods according to the field survey results in Figure 6.

Biodegradable polyesters such as PBAT are moisture sensitive. Thus, the water vapor barrier property is an important item when its application in agricultural mulching film is considered. The WVTR values of each film sampled at different times in the field are presented in Figure 5c. Throughout the whole plating period, the WVTR values of PE always remained below 50 g/m^2^·day. However, at the beginning of the test, the values of these three PBAT-based films were already around 25 times that of the corresponding value of PE film. At the end of the growing season, these data surged to approximately 900 g/m^2^·day to 1000 g/m^2^·day for BDM and BDM1. Due to its serious degradation into pieces, BDM2 could not be tested.

Restraining weeds is also a major function of mulching film. As were shown in Figure 5d, PE film presented the highest percentage of inhibiting weeds, followed by BDM and BDM1. However, BDM2 only reached half the effect of PE, possibly caused by its earlier degradation.

### 3.6. Degradation Behavior

The degradation grading investigation results of each treatment and digital photos of the mulching samples in a fixed square with sides 50 cm long throughout the cotton planting duration are shown in Figure 6. Images taken along a line of mulched cotton are shown in Appendix A. A more detailed and intensive survey result of the qualitative description of the degradation rate is listed in Figure 6a. As shown, with the increase in the mulching time, the three degradable samples gradually degraded, while PE film did not exhibit visible signs of degradation until the mulching ended. Among the degradable films, BDM2 was the first to degrade. As early as 15 days, it showed slight signs of degradation. At 36 days, it has obviously degraded, and abundant weeds were present. BDM and BDM1 started to degrade after two-months. Only a few tears or holes appeared at the beginning but later became increasingly serious and obvious. BDM became fragmented by day 99. The degradation of BDM1 and BDM2 also accelerated further. At harvest time, BDM was almost gone and BDM1 and BDM2 were fragmented. However, the PE was still intact. The results of the degradation investigation of the four samples showed that BDM degraded almost completely.

### 3.7. Effects on Cotton Growth and Yield

The effects of the four films on cotton growth and yield, including emergence rate, canopy cover, plant height, effective boll per plant, and yield, are shown in Figure 7. As shown in Figure 7a, no significant difference was observed among the four treatments for emergency rate, all of which were more than 90%. In terms of the canopy cover, a similar tendency was found as soil temperature changed, and the differences among the four treatments after 70 days of mulching were minimal. At the beginning, the canopy cover of the plants mulched by PE increased significantly. At 57 days, the canopy cover of PE reached almost 60% while all the degradable films were all below 40%. Subsequently, the difference became smaller step by step possibly due to the increasing air temperature and solar radiation. At 70 days, no obvious difference was noted among the crown densities of PE, BMD, and BMD1, but that of BMD2 was approximately10% lower than all the others. The difference disappeared after 84 days of mulching. The plant height and effective boll of each plant showed similarity to the degradation speed of the four samples, which also followed a similar pattern with yield. The differences in plant height were more significant than the effective boll number among the three biodegradable film treatments. The yield of the cotton covered by PE was almost 6000 kg/ha, which was significantly higher than those of degradable films. The yield of cotton covered by BDM2, which was 18.85% lower than that of PE, was also lower than the two other films. These results show that, although the mulching types almost have no effects on the rate of emergence, cotton biomass growth and yield are positively correlated with the time of complete mulching. In this study, the three biodegradable mulch films are not as effective as PE mulch films in promoting the growth of cotton. Because moisture content was the most important environmental variable studied in controlling the polymer’s rate of biodegradation [31], and mulching is widely used in combination with drip-irrigation, further studies for better degradation stability are needed to promote the substitution of BDMs into PE in cotton fields.

## 4. Conclusions

In this study, a pure PBAT film (BDM) and two commercial PBAT-based films (BDM1 and BDM2) with thickness of 10.0 ± 1.0μm were prepared through the same processing of an extrusion blown process to reveal the effects the degradation rate, properties, and structure changes of BDMs along with cotton growing and to evaluate their effects on cotton yield. To compare the effects of the mulch on cotton planting, a traditional commercial PE film was used in a field test with the three PBAT-based ones. The three PBAT-based mulch films were proven to degrade gradually during the mulching period. Field survey results showed that BDM2 began to degrade the earliest followed by BDM and BDM1, whereas BDM degrades faster than BDM1 and BDM2 in later periods of mulching. However, due to the possible occurrence of partial cross-linking during mulching, the microscopic appearance of BDM became smoother, whereby molecular weight, thermal stability, crystallinity, and tensile strength increased slightly. BDM1 and BDM2 showed the opposite change law. With the development of degradation, the WVTR of the three PBAT-based films increased. In fact, when the degradation degree of the BDMs reached Level 3, the measured WVTR values could no longer reflect its moisture retention performance for soil. PE film showed slight cross-linking but no significant changes in its microscopic appearance, thermal performance, barrier properties, or mechanical properties. Field test results showed that PE is most effective in promoting soil temperature, cotton growth, biomass accumulation, as well as in suppressing weeds among all the four films. While these for PBAT-based films are positively correlated with the complete coverage period of the mulching samples. Overall, the biomass accumulation and yield of cotton mulched by BDMs were inferior to those of PE with the same thickness, which were positively correlated with the period of mulching duration. Therefore, in order to improve the yield of cotton covered with BDMs and make it closer to or even equivalent to that of traditional PE film, it is essential to improve its period of mulching duration by means of blending with various polymers, modifying the polyesters, adjusting the processing technology, and increasing the film thickness. Meanwhile, studies concerning the minimum mulching duration of BDMs on cotton planting to ensure its growth and yield are also necessary.

## Figures and Tables

**Figure 1 polymers-14-03157-f001:**
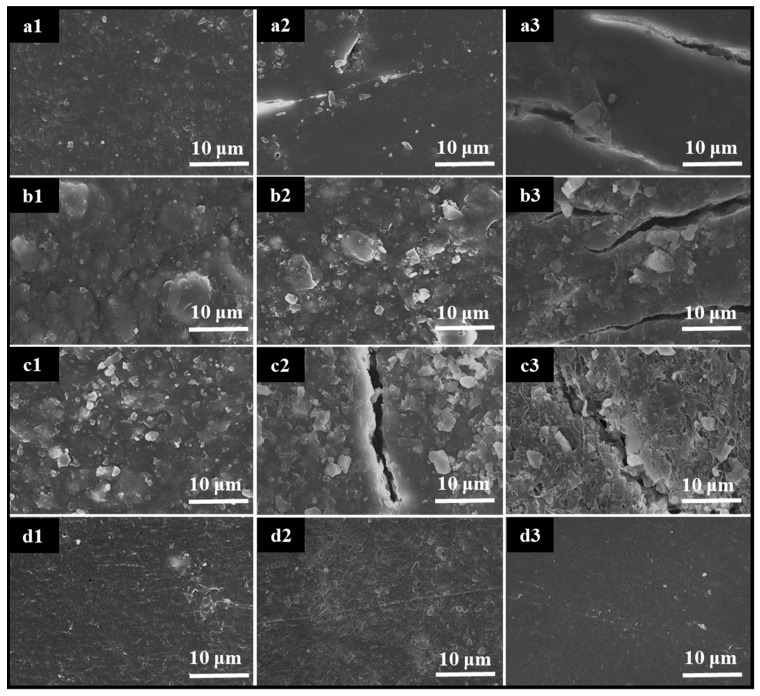
SEM images of BDM (**a**), BDM1 (**b**), BDM2 (**c**), and PE (**d**) after mulching at 0 (**a1**,**b1**,**c1**,**d1**), 87 (**a2**,**b2**,**c2**,**d2**), and 165 (**a3**,**b3,c3**,**d3**) days.

**Figure 2 polymers-14-03157-f002:**
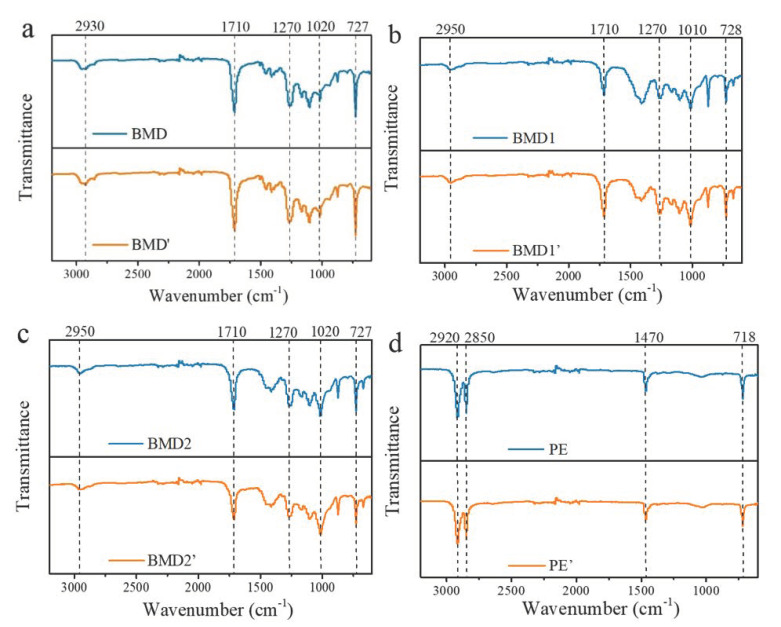
FTIR spectra of films before ((**a**) BDM, (**b**) BDM1, (**c**) BDM2, (**d**) PE) and after mulching ((**a**) BDM’, (**b**) BDM1′, (**c**) BDM2′, (**d**) PE’).

**Figure 3 polymers-14-03157-f003:**
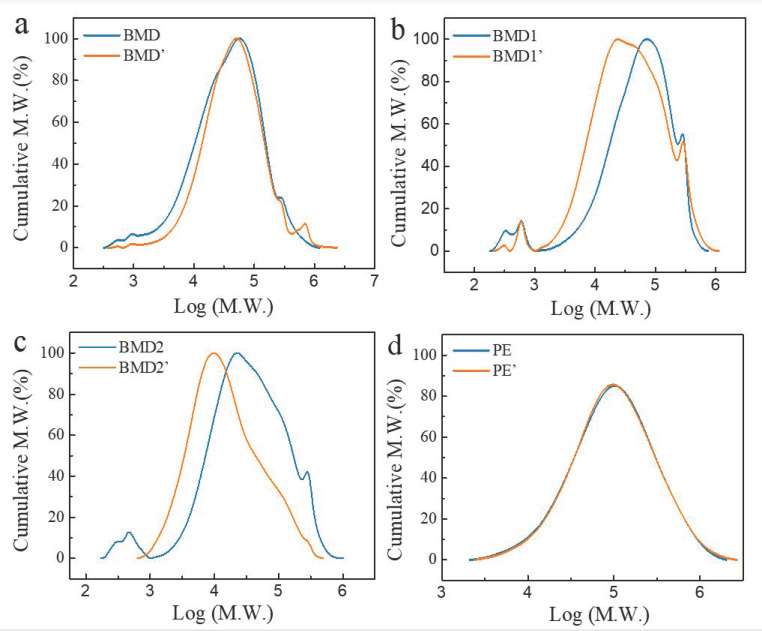
Molecular weight distribution curves of films before ((**a**) BDM, (**b**) BDM1, (**c**) BDM2, (**d**) PE) and after mulching ((**a**) BDM’, (**b**) BDM1′, (**c**) BDM2′, (**d**) PE’).

**Figure 4 polymers-14-03157-f004:**
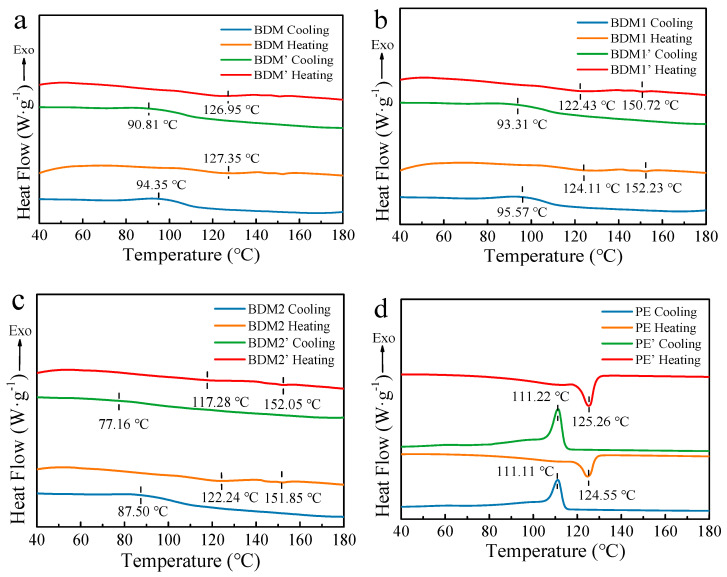
DSC cooling and heating curves of BDM and BDM′ (**a**), BDM1 and BDM1′ (**b**), BDM2 and BDM2′ (**c**), PE and PE′ (**d**).

**Figure 5 polymers-14-03157-f005:**
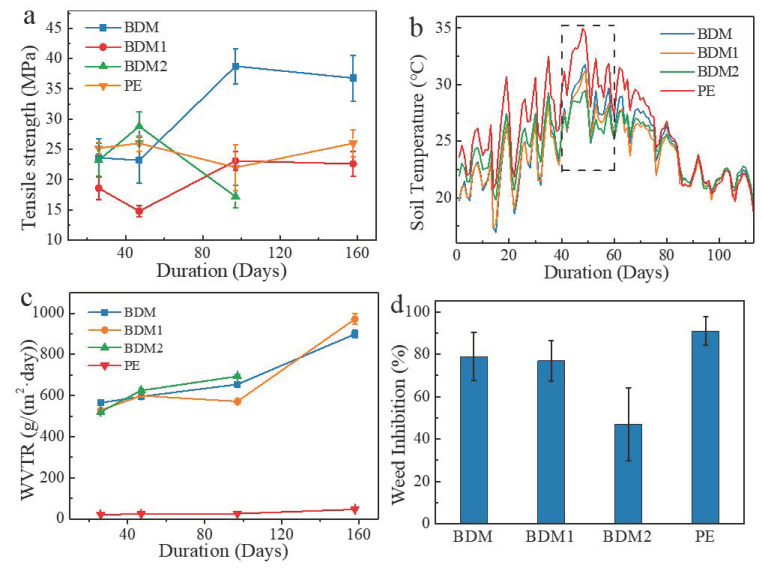
Tensile strength (**a**), soil temperature (**b**), WVTR (**c**) change with mulching duration and weed inhibition capability (**d**) of each treatment.

**Figure 6 polymers-14-03157-f006:**
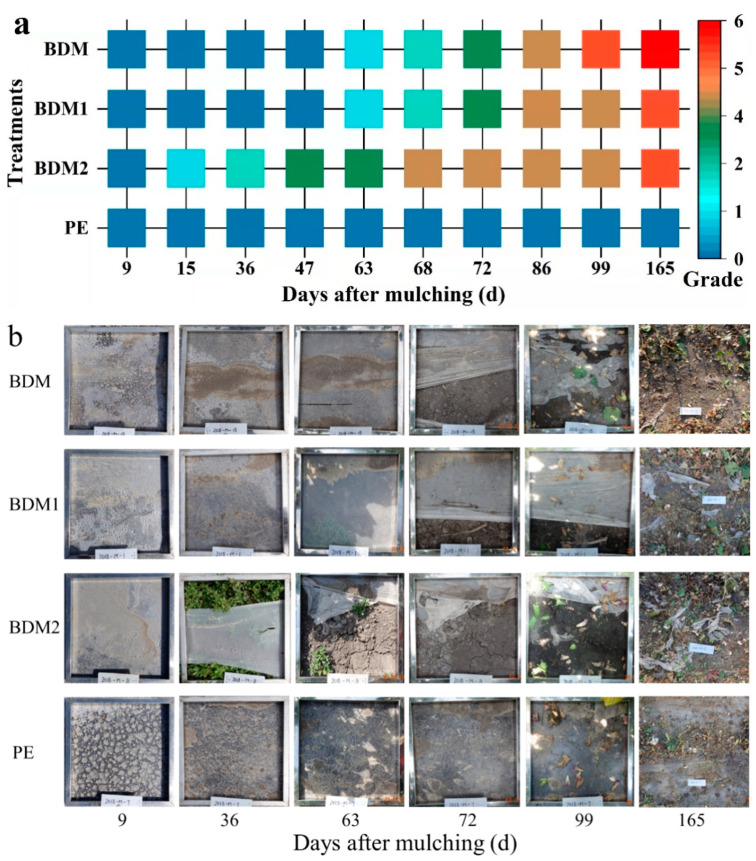
Degradation grading (**a**) and digital photos (**b**) (side length of the metal frame is 50 cm) of BDM, BDM1, BDM2, and PE films along with mulch duration and after harvest (165 days).

**Figure 7 polymers-14-03157-f007:**
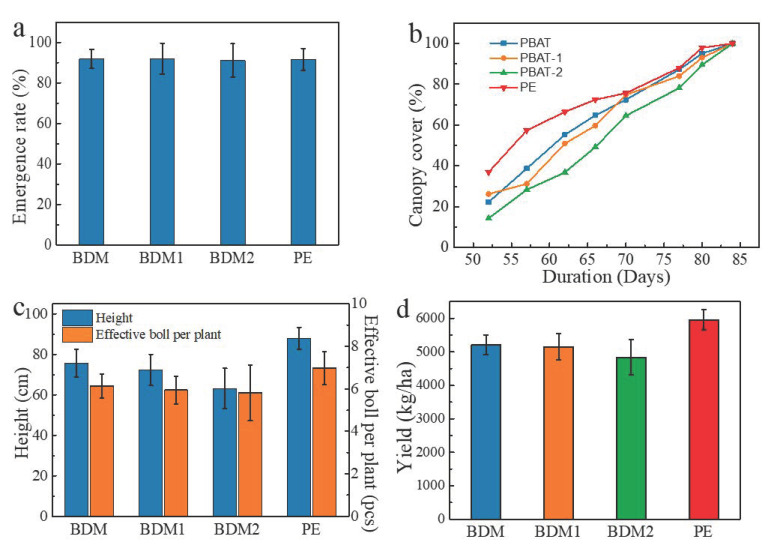
Comparison of emergence rate (**a**), canopy cover (**b**), plant height, boll number (**c**), and cotton yield (**d**) among BDM, BDM1, BDM2, and PE films.

**Table 1 polymers-14-03157-t001:** Quantitative degradation grading observation index.

Level	Holes Number	Holes Diameter (cm)	Crack Length (cm)	Fragment Area (m^2^)
0	0	0	0	-
1	>5	>2	>5	-
2	>5	>10	>50	-
3	>5	>20	>100	-
4	>5	>30	>200	-
5	-	-	-	<0.5
6	-	-	-	<0.05

-: Not applicable.

**Table 2 polymers-14-03157-t002:** Important data from GPC, DSC, and TG determination ^a^.

Samples	GPC			DTG	DSC				
	M_n_ ^b^	M_w_ ^c^	M_w_/M_n_ ^d^	*T*_max_ (°C) ^e^	*T*_c_ (°C)	Δ*H*_c_ (J·g^−1^)	*T*_m_ (°C)	Δ*H*_m_ (J·g^−1^)	*Θ* (%) ^f^
BDM	13,028	69,439	5.33	395.6	94.4	15.2	127.4	11.5	12.1
BDM’	22,841	81,516	3.57	395.8	90.8	14.5	127.0	12.8	
BDM1	8092	88,280	10.9	392.8	95.6	7.77	124.1	3.51	−14.1
BDM1′	11,814	75,028	6.35	391.2	93.3	5.17	122.4	3.01	
BDM2	7571	66,127	8.73	392.4	87.5	7.40	122.2	3.07	−59.3
BDM2′	8063	31,346	3.89	388.0	77.2	4.32	117.3	1.25	
PE	50,975	167,222	3.28	470.2	111.1	99.2	124.6	113.8	0.0
PE’	53,074	172,987	3.26	470.3	111.2	94.9	125.3	113.8	

^a^ The crystallization temperatures (*T*_c_) and the crystallization enthalpy (Δ*H*_c_) were determined from the cooling ramp. The melting temperatures (*T*_m_) and corresponding melting enthalpy (Δ*H*_m_) were obtained from the second heating circle. ^b^ Number average molecular weight. ^c^ Weight average molecular weight. ^d^ Polydispersity index. ^e^ Temperature at the maximum degradation rate, determined by the first derivative thermogravimetric (DTG) curves. ^f^ The percentage of increasing crystallinity, calculated through Equation (2), and in which the melting enthalpy of PBAT were detected in the experiment.

## Data Availability

Not applicable.

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
