# Peer review of "Degradability and Properties of PBAT-Based Biodegradable Mulch Films in Field and Their Effects on Cotton Planting"

_polymers, 2022, doi:10.3390/polym14153157_

Round 1

Reviewer 1 Report

Overall

Plastic mulch management can be a considerable problem for cotton producers when used and refining the characteristics needed from biodegradable alternatives discussed in this paper should be valuable information for parts of the world where the mulch is applied to cotton.  There are some grammar and wording improvements needed as annotated on the manuscript (most of my edits are before section 2.4 and then pick up again after 3.4).  Future studies should include field data on a non-mulched (bare soil) control in addition to the PE film to give an indication when / if the biodegradable material has no impact.  I do not have the material science background to authoritatively comment on the textiles related evaluations, and with that said, the tests used seemed reasonable.  My focus was on the field evaluation methods and results.  Overall, the results seem useful for refining design criteria for biodegradable mulches used in cotton and it was very interesting that the yields were less for the biodegradable mulches tested.  Specific recommendations are included as an attached file in addition to the annotated manuscript. 

Specific

·       Planting rate of 240,000 plants ha−1 is very high for cotton.

·        Section 2.3.3 – it would be helpful if the levels were presented in a table.  For example:

Level

Holes Present

Crack Length

0

0

0

1

5 > 2 cm dia

>  5 cm length

2

5 > 10 cm

> 100 cm

ect

·       Figure 5 is where the data for a bare soil (no mulch) treatment would be most useful.

·       Figure 6 as very informative.

·       Section 3.7 – I believe “canopy cover” is a better term to use than “crown density” in the text and on the y-axis of Figure 7b.

·       After Figure 7, it would be interesting if you could speculate if there were any findings in this study that could improve the performance of biodegradable mulches, and if so, add a statement in the conclusions as well.

Author Response

Reviewer Comment 1:

Plastic mulch management can be a considerable problem for cotton producers when used and refining the characteristics needed from biodegradable alternatives discussed in this paper should be valuable information for parts of the world where the mulch is applied to cotton. There are some grammar and wording improvements needed as annotated on the manuscript (most of my edits are before section 2.4 and then pick up again after 3.4).  Future studies should include field data on a non-mulched (bare soil) control in addition to the PE film to give an indication when / if the biodegradable material has no impact. I do not have the material science background to authoritatively comment on the textiles related evaluations, and with that said, the tests used seemed reasonable. My focus was on the field evaluation methods and results. Overall, the results seem useful for refining design criteria for biodegradable mulches used in cotton and it was very interesting that the yields were less for the biodegradable mulches tested. Specific recommendations are included as an attached file in addition to the annotated manuscript. 

Response:

Thanks very much for all your quite helpful comments and suggestions. We have carefully read them one by one and done corrections and revisions according to those. We also appreciate your specific annotations for grammar and wording improvements you made. We have done the corrections according to base on your detailed notes in the manuscript.

Question1:

Planting rate of 240,000 plants ha−1 is very high for cotton.

Response 1:

Thank you for this comment. The practical planting rate of cotton in the test area is normally 195,000~210,000 plants ha−1. However, some cotton seedlings may be injured or be killed because of various factors during machine seeding process or early growth period. Based on our years of practical experience, the final survival rate of seedlings is about 85%. To ensure the resulting actual planting rate, the theoretical planting rate of 240,000 plants ha−1 was adopted.

Question2: Section 2.3.3 – it would be helpful if the levels were presented in a table.  For example:

Level

Holes Present

Crack Length

0

0

0

1

5 > 2 cm dia

>  5 cm length

2

5 > 10 cm

> 100 cm

ect

Response 2:

Thanks very much for your suggestion. According to this, the quantitative degradation grading observation index were listed in Table 1, which has been added in the manuscript as:

Table 1. Quantitative degradation grading observation index

Level

Holes Number

Holes diameter (cm)

Crack Length (cm)

Fragment area (m2)

0

0

0

0

1

>5 

> 2

>  5

2

>5

> 10

> 50

3

>5

>20

>100

4

>5

>30

>200

5

<0.5

6

<0.05

—: Not applicable

Question 3:

Figure 5 is where the data for a bare soil (no mulch) treatment would be most useful.

Response 3:

Thanks very much for this comment. We would like to explain it as following:

In the present study, we aimed to explore the structural and performance changes of PBAT based biodegradable mulch films compared with PE mulch film. Therefore, the control experiment on a bare soil was not performed in the present work. We are very thankful for your suggestions. We will consider a bare soil (no mulch) treatment when it’s necessary in our other work.

Question 4:

Figure 6 is very informative.

Response 4:

Thanks very much for this comment.

Question 5:

Section 3.7-I believe “canopy cover” is a better term to use than “crown density” in the text and on the y-axis of Figure 7b.

Response 5:

Thanks very much for your suggestion. According to the reviewer’s comment, “Crown density” have been replaced by “Canopy cover” in the text and on the y-axis of Figure 7b. The corresponding description in the manuscript has also been revised.

Question 6:

After Figure 7, it would be interesting if you could speculate if there were any findings in this study that could improve the performance of biodegradable mulches, and if so, add a statement in the conclusions as well.

Response 6:

Thanks very much for your valuable suggestion. According to our research result, the performance and covering duration of BDMs should be promoted for cotton planting use. Based on our other research, the covering period would be longer than around 90 days. However, in this study, we didn’t design any specific new formula or technology for BDMs to test. Therefore, it’s difficult to bring forward a way to improve these. While we have added this statement to the conclusions:

Therefore, in order to improve the yield of cotton covered with BDMs and make it closer to or even reach that of traditional PE film, it is necessary to improve its period of mulching duration through the way of blending with various polymers, modify the polyesters, ad-just the processing technology and increase the film thickness. At the same time, the studies the minimum mulching duration of BDMs on cotton planting are also necessary.

Reviewer 2 Report

In my opinion the paper could become publishable after the following minor revisions will be met:

I. The title and abstract are too long, should be shortened. There exist some editing issues, i.e. "masterbatchof" in section 2.1. 

II. Subsection 2.2. The difference between BDM, BDM1 and BDM2 samples should bespecified in the text. In Figure 2, the x axis is "wavenumber" not "Wave number". Subsection 2.4.3. thermal analysis, the mass and form of the films used in the thermal degradation study should be specified. Figure 4. the direction of the 'exo' or 'endo' should be indicated. Figure S1. TG and DTG curves... on the y axis, replace "weight' with "mass".

Author Response

Reviewer Comment2:

In my opinion the paper could become publishable after the following minor revisions will be met:

Question 1:

The title and abstract are too long, should be shortened. There exist some editing issues, i.e. "masterbatchof" in section 2.1. 

Response 1:

Thanks very much for your comment. According to this comment, the title has been shortened to “Degradation behavior and property changes of PBAT-based biodegradable mulch films in field and their effects on cotton growth and yield”. The abstract has been revised to 239 words as following:

Biodegradable mulches (BDMs) are considered as promising alternative green materials to achieve the substitution of polythene (PE) films to reduce plastic pollution. However, whether the BDMs is effective enough to promote cotton production as PE film is a controversial topic, and comparative studies on structure and properties of BDMs and PE films during mulching in field tests are limited. In this study, laboratory determination and field experiments were conducted with one pure Poly(butylene adipate-co-terephthalate) (PBAT) film (BDM), two commercial PBAT-based films (BDM1 and BDM2), and one PE film to (â…°) compare the degradation behavior, morphology, and property changes of BDMs and PE films  during field application and (â…±) reveal their effects on biomass accumulation and cotton yield. Degradation behavior and , degradation rate, structure, thermal stability, crystallinity, and molecular weight changes of the films before and after mulching were investigated and characterized. The structure, thermal stability, crystallinity, and molecular weight changes of the films before and after mulching were characterized by Fourier transform infrared spectroscopy, thermogravimetric analysis, and gel permeation chromatography. Water vapor transmission rate and mechanical properties of the films and the effects of the films on soil temperature, crop growth, and cotton yield were discussed. Results show that the three PBAT-based mulch films gradually degraded during mulching. The molecular weight, thermal stability, and crystallinity of BDM1 and BDM2 decreased. Interestingly, BDM showed the opposite characteristics, but the degradation degree was greatest at harvest. PE film , which is the most effective for soil temperature accumulation, cotton growth, biomass accumulation, and yield among all samples, showed no significant changes on its microscopic appearance, thermal performance, or properties. These for PBAT-based films were positively correlated with the complete coverage period of the films. In-depth studies that focused on BDMs with longer mulching period must be developed ur-gently to promote the substitution of BDMs into PE to reduce the mulch residual pollution in cotton fields.

And the spelling error of "masterbatchof" has been revised to “master batch for” in part 2.1.

Question 2:

Subsection 2.2. The difference between BDM, BDM1 and BDM2 samples should be specified in the text.

Response 2:

Thanks very much for this helpful comment. The following statements were added to Section 2.2:

“After that the PBAT films were prepared by blown film extruder (QY2200-1, Shandong Plastic Machinery Company, China) at 135 ℃ naming as BDM. Then another two films were prepared through the same blown process conditions directly using commercial PBAT-based master batch Ecoflex® and FLEX-262 F20, which were named as BDM1 and BDM2 respectively. The blow-up ratio of the bubble was 3.5:1. This setting produced a bubble with an average thickness of 10.0±1.0μm (Table S1, see Supplementary Material). "

And the following sentence was added to Section 2.3.2:

The films after mulching in the field were named with BDM', BDM1' and BDM2' correspondingly.

Question 3:

In Figure 2, the x axis is "wavenumber" not "Wave number".

Response 3:

Thanks very much for this comment. According to the reviewer’s comment, "Wave number" have been replaced by "Wavenumber" in Figure 2.

Question 4:

Subsection 2.4.3. thermal analysis, the mass and form of the films used in the thermal degradation study should be specified.

Response 4:

Thanks very much for this comment. The following sentences have been added to Section 2.4.3.

“A total of 5–10 mg sample was tested for films before and after mulching in the field.”

Question 5:

Figure 4. the direction of the 'exo' or 'endo' should be indicated.

Response 5:

Thanks. The direction of the 'Exo' has been indicated in Figure 4.

Question 6:

Figure S1. TG and DTG curves... on the y axis, replace "weight' with "mass".

Response 6:

Thanks for your kindly remind. According to the comment, “Weight” have been replaced by “Mass” on the y axis of TG curves in Figure S1.